

# Long COVID active case detection initiative among COVID-19 patients in Port Dickson, Malaysia: a retrospective study on the positive outcomes, the proportion of patients with long COVID and its associated factors

Kim Sui Wan[1,2], Esther Rishma Sundram[2], Ammar Amsyar Abdul Haddi[2], Abdul Rahman Dashuki[2], Azainorsuzila Ahad[2], Rowena John[2], Muhammad Khairul Ridhuan Abdul Wahid[2], Ungku Izmin Farah Ungku Halmie[2], Farah Edura Ibrahim[2] and Nachia Banu Abdul Rahim[2]

[1] Institute for Public Health, National Institutes of Health, Shah Alam, Selangor, Malaysia
[2] Port Dickson District Health Office, Ministry of Health Malaysia, Port Dickson, Negeri Sembilan, Malaysia

Corresponding author
Kim Sui Wan, kimsui@moh.gov.my, wankimsui@gmail.com

## ABSTRACT

**Background.** Long COVID is new or ongoing symptoms at four weeks or more after the start of acute COVID-19. However, the prevalence and factors associated with long COVID are largely unknown in Malaysia. We aim to determine the proportion and factors associated with long COVID among COVID-19 patients in Port Dickson, Malaysia. The positive outcomes of our long COVID active detection initiative were also described.

**Methods.** This was a retrospective analysis of long COVID data collected by the Port Dickson District Health Office between 1 September 2021 to 31 October 2021. Monitoring long COVID symptoms was our quality improvement initiative to safeguard residents' health in the district. The study population was patients previously diagnosed with COVID-19 who resided in Port Dickson. The inclusion criteria were adults aged 18 years and above and were in the fifth week (day 29 to 35) post-COVID-19 diagnosis during the data collection period. We called all consecutive eligible patients to inquire regarding long COVID symptoms. Long COVID was defined as new or ongoing symptoms lasting more than 28 days from the date of positive SARS-CoV-2 by polymerase chain reaction test. Binary multivariate logistic regression was conducted to determine factors associated with long COVID.

**Results.** Among 452 patients, they were predominantly male (54.2%), Malays (68.8%) and aged 18–29 years (58.6%). A total of 27.4% (95% CI [23.4–31.8]) of patients experienced long COVID symptoms and were referred to government clinics. The most frequent long COVID symptoms experienced were fatigue (54.0%), cough (20.2%), muscle pain (18.5%), headache (17.7%) and sleep disturbance (16.1%). Females, patients with underlying cardiovascular disease, asthma and chronic obstructive airway disease, those who received symptomatic care, and patients with myalgia and headaches at COVID-19 diagnosis were more likely to have long COVID. Three patients with

suspected severe mental health problems were referred to the district psychologist, and ten patients with no/incomplete vaccination were referred for vaccination.
**Conclusion**. Long COVID is highly prevalent among COVID-19 patients in Port Dickson, Malaysia. Long-term surveillance and management of long COVID, especially among the high-risk groups, are needed as we transition to living with COVID-19.

# INTRODUCTION

The novel coronavirus SARS-CoV-2 has paved its way into eventually causing the biggest pandemic in the 21st century with almost 520 million coronavirus disease 19 (COVID-19) cases and over 6.1 million deaths worldwide as of 17 May 2022 (*World Health Organisation, 2022b*). Evidence has emerged as the pandemic progresses that some patients are experiencing prolonged multiorgan symptoms and complications beyond the initial period of acute infection and illness (*Venkatesan, 2021*). The new or ongoing symptoms four weeks or more after the start of acute COVID-19 are often described as long COVID (*National Institute for Health Care Excellence, 2020*). Long COVID is classified into ongoing symptomatic COVID-19 and post-COVID-19 syndrome. Ongoing symptomatic COVID-19 refers to signs and symptoms of COVID-19 from four to twelve weeks (*Ministry of Health Malaysia, 2021e*; *National Institute for Health Care Excellence, 2020*). Meanwhile, post-COVID-19 syndrome means signs and symptoms that develop during or after an infection consistent with COVID-19, continue for more than 12 weeks and are not explained by an alternative diagnosis (*Ministry of Health Malaysia, 2021e*; *National Institute for Health Care Excellence, 2020*).

The long COVID symptoms are highly variable and wide-ranging from respiratory, cardiovascular, neurological, gastrointestinal, musculoskeletal, psychological, ear, nose and throat, dermatological and generalised symptoms (*Ministry of Health Malaysia, 2021e*; *National Institute for Health Care Excellence, 2020*). In the United Kingdom, around one in five people with COVID-19 had symptoms lasting for ≥5 weeks, and one in ten had symptoms lasting for ≥12 weeks (*Office for National Statistics United Kingdom, 2020*). In a prospective cohort study of COVID-19 symptoms, 13.3% of COVID-19 cases had symptoms lasting >28 days, 4.5% for >8 weeks, and 2.3% for >12 weeks (*Sudre et al., 2021*). The long COVID prevalence equated to an estimated 186,000 individuals (95% CI [153,000–221,000]) in England who had symptoms persisting between 5 and 12 weeks (*Venkatesan, 2021*). The high number implied that many resources were needed to help patients and clinicians to understand and manage the long-term effects of COVID-19 (*Venkatesan, 2021*).

Malaysia, an upper-middle-income country, was not spared from the pandemic and had over 4.4 million cases and 35,620 deaths as of 17 May 2022 (*World Health Organisation, 2022a*). An online survey in Malaysia reported that 21% of COVID-19 patients had

long COVID, and females and those with more severe acute illnesses had higher odds of experiencing long COVID (*Moy et al., 2022*). While the study is the first in Malaysia to describe long COVID and its associated factors, selection bias could be present due to the online method in data collection; hence, more local studies are required (*Moy et al., 2022*).

Besides the study, the prevalence and factors associated with long COVID symptoms are largely unknown locally. A conceptual framework has been proposed to guide identifying and mitigating long COVID predictors (*Kondratiuk et al., 2021*). Understanding the characteristics of susceptible individuals, acute predictive markers, and convalescent predictive markers will assist public health practitioners and clinicians in planning and managing long COVID (*Kondratiuk et al., 2021*). As a part of our ongoing quality improvement initiative to safeguard the health of residents in Port Dickson, we started monitoring long COVID symptoms among COVID-19 patients. This study uses real-world data to determine the proportion and factors associated with long COVID among COVID-19 patients in Port Dickson, Malaysia. We further describe the additional positive outcomes of the active detection initiative of long COVID.

## MATERIALS & METHODS

### Study design and setting

This was a secondary data analysis of the routine long COVID information collected by the Port Dickson District Health Office between 1 September 2021 to 31 October 2021. The data source came from the long COVID active case detection project, one of the quality programmes in our district health office that aims to identify and refer patients with long COVID to nearby government health clinics. In other words, this public health service is unique to Port Dickson District Health Office.

Port Dickson district is one of the seven districts in Negeri Sembilan state, Malaysia. The district is located about 90 km south of Kuala Lumpur, the capital city of Malaysia, with an estimated population of 128,689 people in 2020 (*Department of Statistics Malaysia, 2022*). The study population was patients previously diagnosed with COVID-19 who resided in Port Dickson. The inclusion criteria were adults aged 18 years and above and were in the fifth week (day 29 to 35) post-COVID-19 diagnosis during the data collection period. We excluded those who have died.

### Data collection flow

The Port Dickson District Health Office is responsible for healthcare delivery and public health services in the district alongside six government healthcare clinics and six smaller rural/community clinics. The critical functional unit during the current COVID-19 pandemic is the Port Dickson Crisis Preparedness and Response Centre, which leads and coordinates all prevention and control activities.

The centre continuously received notification of new COVID-19 cases through three main channels: (i) the State Health Department of Negeri Sembilan *via* the National Public Health Laboratory Information System, (ii) public and private healthcare facilities, and (iii) COVID-19 Assessment Centres. We defined a COVID-19 patient as a person with a laboratory confirmation for SARS-CoV-2 by reverse transcriptase-polymerase chain

reaction (RT-PCR) test (*Ministry of Health Malaysia, 2021b*). It is mandatory in Malaysia for every medical practitioner to notify infectious diseases, including COVID-19 (*Laws of Malaysia, 1988*). Our team compiled and collated the notifications in a line-listing format on a daily basis.

A team of eight medical doctors was trained for this long COVID active case detection initiative. We telephoned patients to interview them about long COVID symptoms in the fifth week (day 29 to 35) post-COVID-19 diagnosis. The interviews were guided by a data collection spreadsheet displayed on computers. The interviewer recorded answers that corresponded with the pre-coded responses. When our telephone calls were unanswered three times on different days, we labelled them as a loss of contact.

When patients were identified to have long COVID, we referred them to the nearest government health clinics. For patients with psychological symptoms, our doctors assessed their clinical history and referred patients to the district psychologist. Besides that, we identified and referred patients with no or incomplete COVID-19 vaccination to the vaccine administration centre.

## Dependent variable

The primary outcome of this study was to portray the proportion of COVID-19 patients that experienced any long COVID symptoms. We defined long COVID as new or ongoing symptoms lasting more than 28 days from the date of positive SARS-CoV-2 by RT-PCR (*National Institute for Health Care Excellence, 2020*). The categories of long COVID symptoms were: (a) respiratory symptoms—breathlessness and cough; (b) cardiovascular symptoms—chest tightness, chest pain and palpitations; (c) generalised symptoms—fatigue and fever; (d) neurological symptoms—loss of concentration, memory impairment, headache, sleep disturbance and dizziness; (e) gastrointestinal symptoms—abdominal pain, nausea and diarrhoea; (f) musculoskeletal symptoms—joint pain and muscle pain; (g) psychological symptoms—depression, anxiety and stress symptoms; and (h) ear, nose, throat symptoms—tinnitus, earache, sore throat, dizziness, loss of taste and loss of smell (*National Institute for Health Care Excellence, 2020*).

## Independent variables

Patients self-reported their ages, sexes, ethnicities, smoking status, body weights, past medical histories, vaccination statuses, acute COVID-19 symptoms, the maximum care received, and the management settings during acute illnesses. These data formed the baseline characteristics and independent variables in this analysis. The ethnic groups were Malays, Chinese, Indians, and other ethnicities in this multiracial country. Smoking status was yes or no to current smoking. Obesity was operationally defined as self-reported body weight above 90 kg. Body mass index was not used because patients often did not know their heights, and physical measurement was not possible during phone conversations. Our Ministry of Health used the body mass index cut-off of 35 kg/m$^2$ as one of the criteria for admission to the Low-Risk COVID-19 Quarantine and Treatment Centre (*Ministry of Health Malaysia, 2021c*). Hence, based on the nation's estimated mean height and practical reasons, we defined obesity as weight above 90 kg (*NCD Risk Factor Collaboration, 2016*).

The comorbidities included diabetes, hypertension, cardiovascular disease (CVD), asthma, chronic obstructive airway disease (COAD), neoplasm, thyroid disease, anaemia, gastritis and spondylitis. The vaccine types received by patients during this period were either Comirnaty (Pfizer-BioNTech, New York, NY, USA) or CoronaVac (Sinovac, Beijing, China). The COVID-19 symptoms during acute illnesses were fever, cough, general weakness or fatigue, headache, myalgia, sore throat, coryza, dyspnoea, anorexia, nausea or vomiting, diarrhoea, loss of taste, and loss of smell (*Ministry of Health Malaysia, 2021a*).

We categorised the maximum care received by patients during their acute COVID-19 illnesses into (i) asymptomatic, (ii) symptomatic but did not require oxygen therapy, (iii) symptomatic and required oxygen therapy, and (iv) intubation (*Ministry of Health Malaysia, 2021a*; *Ministry of Health Malaysia, 2021b*). All confirmed cases of COVID-19 had to be isolated at homes, Low-Risk COVID-19 Quarantine and Treatment Centres, or hospitals, depending on their ages, symptoms and comorbidities (*Ministry of Health Malaysia, 2021d*). Adults below 60 years old, with no or only mild symptoms, without comorbidities, or with known controlled comorbidities were isolated at homes (*Ministry of Health Malaysia, 2021d*). For paediatric age groups, home isolation was suitable for those without symptoms, who had no comorbidity and with appropriate caregivers (*Ministry of Health Malaysia, 2021d*).

The criteria for admission to the Low-Risk COVID-19 Quarantine and Treatment Centre were: (a) patients who fulfilled home monitoring criteria but without suitable home conditions; (b) could ambulate without assistance and self-administered medications and (c) did not have ongoing clinical needs such as haemodialysis (*Ministry of Health Malaysia, 2021d*). Hospital admissions were for patients with: (a) moderate to severe clinical stages; (b) age above 60 years; (c) all paediatric cases with comorbidity or those below two years old; (d) uncontrolled comorbidity such as having unstable angina; (e) end-stage renal failure on dialysis and (f) pregnant mothers (*Ministry of Health Malaysia, 2021d*).

## Sample size

We estimated the sample size using online OpenEpi (version 3.01) sample size for frequency in a population (*Dean, Sullivan & Soe, 2013*). Using an estimated long COVID prevalence of 20%, as reported in Malaysia and the United Kingdom, in the Port Dickson district with a finite population of 128,689 and a 95% confidence level, the minimum sample size was 246 (*Department of Statistics Malaysia, 2022*; *Office for National Statistics United Kingdom, 2020*).

## Statistical analyses

Descriptive analysis was performed for the baseline characteristics and long COVID symptoms. For normally distributed data, the variables were presented in mean ± standard deviation (SD), while median and inter-quartile range (IQR) were presented for skewed data. Both frequency and percentages were reported for categorical variables.

Univariate binary logistic regression was first conducted to determine individual factors associated with long COVID symptoms. Multivariate logistic regression analysis was carried out for variables with $P < 0.25$ and clinically essential variables. The Omnibus test of model

coefficients was used to check the model against the baseline model. The coefficient of determination, $R^2$, represented the amount of variation in the outcome that the model explained. The Hosmer & Lemeshow test indicated if the model fit the data well. The model's predictive power was assessed using the classification table and area under the Receiver Operating Characteristic (ROC) curve. We reported the adjusted odd ratios with 95% confidence intervals. All the tests were two-tailed and statistical significance was fixed at $P < 0.05$. All analyses were done using the IBM SPSS Statistical Software, version 23.

## Ethical approval

This paper utilised anonymised secondary data from Port Dickson District Health Office. All patients' identifiers, such as names, identity card numbers, contact numbers, and addresses, were first removed. The Medical Research Ethics Committee (MREC) Ministry of Health Malaysia approved this study (NMRR ID-22-01103-GRF). The MREC waived written informed consent following local legislation and national guidelines because this study was a retrospective analysis of an anonymised dataset.

## RESULTS

A total of 452 COVID-19 cases were analysed, and patients were predominantly male (54.2%), Malays (68.8%) and were in the 18-29-year age category (58.6%) (Table 1). It was observed that 28.3% of patients were smokers, and most cases (73.2%) did not have comorbidities. Among those with comorbidities, the most common comorbidities reported were obesity (45.5%), followed by hypertension (36.3%) and diabetes (26.4%). Additionally, 77.2% of cases reported having COVID-19 symptoms at the time of diagnosis. The most common symptoms experienced were fever (73.6%) followed by cough (63.3%) and loss of taste or ageusia (57.9%). More than half of the patients (55.3%) received two doses of COVID-19 vaccines at the time of COVID-19 diagnosis, whereas only 20.1% did not receive any vaccination dose. Most patients were treated at the Low-risk Quarantine and Treatment Centre (63.3%) after diagnosis, and the maximum care received was mainly symptomatic treatment and did not require oxygen supplementation (73.2%).

Table 2 exhibits the descriptive characteristics of long COVID symptoms reported in the fifth-week post-diagnosis. We discovered that 124 or 27.4% (95% CI [23.4–31.8]) of patients experienced long COVID symptoms. The most frequent symptoms experienced were fatigue (54.0%), cough (20.2%), muscle pain (18.5%), headache (17.7%) and sleep disturbance (16.1%). In contrast, the least common long COVID symptoms were diarrhoea (0.0%), earache (0.8%), abdominal pain (1.6%), and fever (1.6%).

Simple logistic regression was performed, and variables with $P$ values <0.25 were selected to be included in the final model. These variables were age group, sex, ethnicity, smoking status, obesity, cardiovascular disease, asthma and COAD, fever, cough, fatigue, headache, myalgia, sore throat, coryza, dyspnoea, anorexia, nausea, or vomiting, diarrhoea, loss of taste and loss of smell (Table S1).

In the final multiple logistic regression model, six independent factors were associated with long COVID: females, underlying cardiovascular disease, asthma/COAD, headaches, myalgia and receiving symptomatic treatment but not oxygen (Table 3). Underlying

**Table 1** Baseline characteristics of COVID-19 patients.

| Baseline characteristics | n = 452 | % (100.0) |
|---|---|---|
| **Age** | | |
| Years, median (IQR) | 28.0 (11.0) | |
| 18–29 | 265 | 58.6 |
| 30–39 | 101 | 22.3 |
| 40–49 | 36 | 8.0 |
| 50–59 | 24 | 5.3 |
| ≥ 60 | 26 | 5.8 |
| **Sex** | | |
| Male | 245 | 54.2 |
| Female | 207 | 45.8 |
| **Ethnicity** | | |
| Malay | 311 | 68.8 |
| Chinese | 46 | 10.2 |
| Indian | 82 | 18.1 |
| Others | 13 | 2.9 |
| **Smoking** | 128 | 28.3 |
| **Comorbidities** | | |
| No | 331 | 73.2 |
| Yes | 121 | 26.8 |
| Weight >90 kg (Obesity) | 55 | 45.5 |
| Diabetes | 32 | 26.4 |
| Hypertension | 44 | 36.3 |
| Cardiovascular disease | 9 | 7.4 |
| Asthma and COAD | 14 | 11.6 |
| Neoplasm | 2 | 1.7 |
| Thyroid disease | 3 | 2.5 |
| Others: anaemia, gastritis, spondylitis | 13 | 10.7 |
| **COVID-19 symptoms at onset** | | |
| No | 103 | 22.8 |
| Yes | 349 | 77.2 |
| Fever | 257 | 73.6 |
| Cough | 221 | 63.3 |
| General weakness or fatigue | 151 | 43.3 |
| Headache | 128 | 36.7 |
| Myalgia | 132 | 37.8 |
| Sore throat | 109 | 31.2 |
| Coryza | 161 | 46.1 |
| Dyspnoea | 45 | 12.9 |
| Anorexia, nausea or vomiting | 24 | 6.9 |
| Diarrhoea | 53 | 15.2 |
| Loss of taste | 170 | 48.7 |
| Loss of smell | 202 | 57.9 |
| **COVID-19 vaccination status during diagnosis** | | |
| Two doses of Comirnaty or CoronaVac | 250 | 55.3 |
| Only one dose of Comirnaty or CoronaVac | 111 | 24.6 |
| Not vaccinated | 91 | 20.1 |

**Table 1** (*continued*)

| Baseline characteristics | *n* = 452 | % (100.0) |
|---|---|---|
| **Management settings** | | |
| Home | 109 | 24.1 |
| Low-risk Quarantine and Treatment Centre | 286 | 63.3 |
| Hospital | 57 | 12.6 |
| **Maximum care received** | | |
| Asymptomatic | 104 | 23.0 |
| Symptomatic, not requiring oxygen supplement | 331 | 73.2 |
| Symptomatic, requiring oxygen supplement | 16 | 3.5 |
| Intubation | 1 | 0.2 |

**Notes.**
COAD, Chronic obstructive airway disease.

cardiovascular disease yielded the highest odds (aOR: 20.8, 95% CI [3.84–113.2]). The second strongest associated factor was asthma and COAD (aOR:3.39, 95% CI [1.09–10.58]). This was followed by patients receiving symptomatic care without oxygen treatment (aOR:2.28, 95% CI [1.15–4.51]). Females, patients with myalgia, and those with headaches at COVID-19 diagnosis were twice as likely to develop long COVID compared to their respective counterparts.

We referred all 124 patients with long COVID to the nearest government clinics for further assessment by medical doctors. Three patients with suspected severe mental health problems were referred to the district psychologist for further intervention. In addition, ten patients were identified to have no/incomplete COVID-19 vaccination and were referred to the vaccine administration centre for vaccination.

## DISCUSSION

### Proportion of patients with long COVID

The proportion of our patients with long COVID was within the wide range reported in a systematic review (*Cabrera Martimbianco et al., 2021*). The prevalence varied widely due to differences in population, the accuracy of diagnosis, the capability of the healthcare system, the duration of follow-up, the reporting systems and the symptoms examined (*Cabrera Martimbianco et al., 2021*). Our top five long COVID symptoms had similarities with preliminary findings from the Ministry of Health, Malaysia research on long COVID patients: lethargy, breathing difficulties when performing certain tasks, coughing, insomnia, and anxiety (*Carvalho, 2021*). Besides that, a recently published local study reported fatigue, brain fog, arthralgia/myalgia, and insomnia as the most common long COVID symptoms at up to six weeks which were quite similar to ours (*Moy et al., 2022*). The similarities imply that our results were consistent with other local studies.

### Long COVID symptoms

Our most common long COVID symptom of fatigue is similar to many studies (*Cabrera Martimbianco et al., 2021*; *Crook et al., 2021*; *D'Cruz et al., 2021*; *Moy et al., 2022*). Evidence shows that miscommunication in the inflammatory response pathways may result in chronic fatigue (*Islam, Cotler & Jason, 2020*). Olfactory neuron damage leads to increased resistance to cerebrospinal fluid drainage through the cribriform plate, and this

**Table 2  Long COVID symptoms at fifth week post diagnosis ($n = 452$).**

| Long COVID symptoms | n | % |
|---|---|---|
| **Asymptomatic** | 328 | 72.6 |
| **Symptomatic** | 124 | 27.4 |
| Respiratory symptoms | | |
| - Cough | 25 | 20.2 |
| - Breathlessness | 18 | 14.5 |
| Cardiovascular symptoms | | |
| - Chest tightness/pain | 12 | 9.7 |
| - Palpitation | 7 | 5.6 |
| Generalised symptoms | | |
| - Fatigue | 67 | 54.0 |
| - Fever | 2 | 1.6 |
| Neurological symptoms | | |
| - Loss of concentration | 13 | 10.5 |
| - Memory impairment | 16 | 12.9 |
| - Headache | 22 | 17.7 |
| - Sleep disturbance | 20 | 16.1 |
| - Dizziness | 14 | 11.3 |
| Gastrointestinal symptoms | | |
| - Abdominal pain | 2 | 1.6 |
| - Nausea | 3 | 2.4 |
| - Diarrhoea | 0 | 0.0 |
| Musculoskeletal symptoms | | |
| - Joint pain | 14 | 11.3 |
| - Muscle pain | 23 | 18.5 |
| Psychological symptoms | | |
| - Depression symptoms | 3 | 2.4 |
| - Anxiety symptoms | 8 | 6.5 |
| - Stress symptoms | 5 | 4.0 |
| Ear, nose, and throat symptoms | | |
| - Tinnitus | 6 | 4.8 |
| - Earache | 1 | 0.8 |
| - Sore throat | 8 | 6.5 |
| - Loss of taste | 3 | 2.4 |
| - Loss of smell | 7 | 5.6 |

**Notes.**
The cumulative total for all symptoms could exceed the total number of patients as each patient might have multiple long COVID symptoms.

indirectly affects the central nervous system by congesting the glymphatic system (*Jessen et al., 2015*). When the glymphatic system is congested, toxin accumulation may lead to post COVID-19 fatigue (*Wostyn, 2021*).

The cough frequency among our long COVID patients was consistent with the prevalence range reported in a systematic review (*Cabrera Martimbianco et al., 2021*). The pathophysiology of cough is hypothesised to be due to a neuroinflammatory response

**Table 3  Independent factors associated with long COVID.**

| Baseline characteristics | Adjusted OR | 95% CI for adjusted OR | *P* values |
|---|---|---|---|
| Female | 2.09 | 1.33–3.29 | 0.001 |
| Cardiovascular disease | 20.84 | 3.84–113.2 | <0.001 |
| Asthma and COAD | 3.39 | 1.09–10.58 | 0.035 |
| Headache | 1.97 | 1.20–3.24 | 0.008 |
| Myalgia | 1.98 | 1.20–3.27 | 0.007 |
| Maximum care received | | | |
| - Asymptomatic | 1.00 | | |
| - Symptomatic, not requiring oxygen | 2.28 | 1.15–4.51 | 0.019 |
| - Symptomatic, requiring oxygen/intubation | 1.30 | 0.34–4.92 | 0.705 |

Notes.
COAD, Chronic obstructive airway disease.
Omnibus tests of model coefficients ($P < 0.001$).
Hosmer-Lemeshow goodness of fit test ($P = 0.930$), classification table (overall correct percentage: 75.7%), −2 log likelihood (466.5), Nagelkerke R square (0.19).
Area under receiving operating characteristics, ROC curve: 0.73, standard error 0.03, $P < 0.001$ and 95% CI [0.68–0.78].

or SARS-CoV-2 invasion to vagal sensory nerves causing hypersensitivity of the cough pathways (*Song et al., 2021*). Besides that, the neuroinflammatory response also affects various regions in the brain to induce symptoms like pain, headache, sleep disturbance and others (*Song et al., 2021*). This mechanism may explain the long COVID symptoms among our patients.

The relatively low prevalence of fever, gastrointestinal, and ear, nose and throat symptoms as long COVID symptoms were quite similar to those reported in a systematic review and local Malaysian study (*Cabrera Martimbianco et al., 2021*; *Moy et al., 2022*). The broad frequency variation of long COVID symptoms highlights the urgent need to understand this emerging, complex, and challenging medical condition (*Cabrera Martimbianco et al., 2021*).

## Factors associated with long COVID

Our analysis revealed that patients with underlying cardiovascular disease (CVD) is the strongest factor associated with long COVID. This could be a new finding because limited studies show direct associations between underlying CVD with long COVID. Pre-existing CVD did not show significant associations with long COVID (*Crook et al., 2021*; *Moy et al., 2022*). Nevertheless, this result should be interpreted with caution due to the small sample size of patients with underlying CVDs and the wide 95% confidence intervals.

Previous evidence has indicated that underlying health conditions, including CVD and its risk factors, may lead to severe COVID-19 and mortality (*Matsushita et al., 2020*; *Yang et al., 2020*). A systematic review and meta-analysis reported that people with prior CVD had a pooled relative risk of 5.05 for severe COVID-19 (*Matsushita et al., 2020*). CVD as a risk factor for severe COVID-19 may partially explain the association between underlying CVD with long COVID. Patients recovering from severe COVID-19 may require an extended period to recover, contributing to the persistence of symptoms beyond 28 days. The severity of long COVID is correlated with the severity of acute COVID-19 infection (*Kamal et al.,*

*2021*). It was observed that patients with more severe acute COVID-19 were six times more likely to develop severe long COVID (*Menezes Jr et al., 2022*).

Besides that, we hypothesise that cardiac damages following COVID-19 may worsen in patients with CVD who later develop long COVID. COVID-19 is linked with the new onset of cardiovascular complications (*Harrison et al., 2021*). Two cohort studies reported that COVID-19 survivors had increased risks of incident CVD spanning several categories, including ischemic and non-ischemic heart disease, pericarditis, myocarditis, heart failure, cerebrovascular disorder, and thromboembolic disease (*Wang et al., 2022*; *Xie et al., 2022*). Long-term effects of myocardial inflammation in recovered COVID-19 patients have also been reported (*Puntmann et al., 2020*).

Our result that females were more likely to have long COVID was consistent with other studies (*Bai et al., 2022*; *Moy et al., 2022*; *Thompson et al., 2022*; *Torjesen, 2021*). Middle-aged women have a higher risk of experiencing a range of debilitating ongoing symptoms, such as fatigue, breathlessness, muscle pain, anxiety, depression, and 'brain fog' after hospital treatment for COVID-19 (*Torjesen, 2021*). Hormonal differences may result in hyperinflammatory status among females in the early phase of the disease (*Bienvenu et al., 2020*). A stronger IgG antibodies production among females at the beginning of COVID-19 could lead to different disease outcomes between sexes (*Zeng et al., 2020*).

The primary target organ for COVID-19 is the respiratory tract. Hence, our finding of an association between underlying asthma with long COVID symptoms is not surprising and has been similarly reported (*Michelen et al., 2021*; *Sudre et al., 2021*; *Thompson et al., 2022*). Substantial lung and respiratory tract damage can occur as SARS-CoV-2 replicates inside endothelial cells, resulting in endothelial damage and intense immune and inflammatory reaction (*Kempuraj et al., 2020*). Patients with pre-existing lung abnormalities are more likely to develop fibrotic-like changes to lung tissue (*Han et al., 2021*). Thus, underlying asthma and COAD may be a precipitating factor in long COVID symptoms.

We found that patients who presented with headaches at the onset of COVID-19 diagnosis are three times more likely to develop long COVID symptoms. Although headaches are among the most frequent neurological symptoms of long COVID, the exact causes of headaches in long COVID remain uncertain (*Martelletti et al., 2021*). Possible mechanisms are direct neuro-invasion with damage to the neuronal pathway and indirect effects mediated by hypoxia, hypertension, coagulopathy and cytokine storm on the central nervous system (*Martelletti et al., 2021*). The worsening of pre-existing brain diseases and the development of new ones, such as cerebrovascular events, infectious and toxic encephalopathy and meningoencephalitis, may also explain the symptom (*Martelletti et al., 2021*).

Patients with any viral infections commonly experience myalgia. However, myalgia caused by COVID-19 infection persists longer and is more severe than the usual myalgia of other viral infections. Generalised inflammation and cytokine response inside the musculoskeletal system can cause myalgia (*Henry, Lippi & Wong, 2020*). In COVID-19, SARS-CoV-2 can penetrate ACE2 receptors, causing infection and damage to the muscle (*Kucuk, Cumhur Cure & Cure, 2020*). Myalgia will be persistent, and long COVID symptoms will develop due to the increased lactate levels, low pH, and low oxygen levels

(*Kucuk, Cumhur Cure & Cure, 2020*). The evidence supported the results of our analysis that patients with myalgia at the time of diagnosis were more prone to develop long COVID symptoms.

Our symptomatic patients at disease onset were more likely to develop long COVID than asymptomatic ones. Symptomatic disease is a proxy for more severe COVID-19 and can explain its association with long COVID. Previous studies reported that patients hospitalised for severe COVID-19 frequently suffer long-term symptoms (*Bellan et al., 2021*; *Huang et al., 2021*). For instance, patients who were more severely ill during their hospital stay had worse pulmonary functions six months after the acute infection (*Huang et al., 2021*). In Malaysia, patients with moderate and severe acute COVID-19 symptoms were more likely to develop long COVID (*Moy et al., 2022*). A systematic review concluded that severe clinical status is a risk factor for long COVID (*Cabrera Martimbianco et al., 2021*).

### Positive outcomes of long COVID active detection initiative

Our initiative to actively detect long COVID has yielded some positive outcomes. Since the natural progression of long COVID remains uncertain, there is a need to investigate, treat, and follow up with the patients (*Ministry of Health Malaysia, 2021e*). Hence, we referred all patients with long COVID symptoms to the nearest government clinics for multidisciplinary team management (*Ministry of Health Malaysia, 2021e*). At the time of our initiative, the COVID-19 vaccination coverage in Port Dickson had achieved around 90%, and we had observed a decrease in daily vaccine uptake. Through our long COVID active case detection, we managed to identify and refer ex-COVID-19 patients who were yet to be vaccinated.

A study reported a high prevalence of depression, anxiety, and stress symptoms in Malaysia during this COVID-19 pandemic (*Tay et al., 2022*). Evidence has shown that patients with COVID-19 have higher risks of mental health problems several months post-initial infection (*Huang et al., 2021*; *Taquet et al., 2021*). The reasons are multifactorial and may include the direct effect of SARS-CoV-2 infection, the immunological response, corticosteroid treatment, stay in intensive care units (ICU), social isolation, and stigma (*Huang et al., 2021*; *Rogers et al., 2020*). After elaborate phone call conversations with patients, our medical doctors identified and referred three patients suspected of severe mental health problems to the psychologist.

### Strengths and limitations

The main strength of this analysis is the use of real-world data to determine the proportion of COVID-19 patients with long COVID and its associated factors. Our findings can add new knowledge to the pool of information on long COVID, especially in Malaysia. Implementing this long COVID surveillance initiative helped improve the quality of public health service and may reduce the long-term disease burden posed by long COVID among COVID-19 patients in our district.

We acknowledge several limitations. Firstly, patients were asked to recall their symptoms and experience several weeks after their acute illness and thus might be subjected to

recall bias. Secondly, the demographic characteristics of our COVID patients were not representative of the population in Port Dickson and Malaysia. These differences were not unexpected because higher proportions of COVID-19 patients were reported during the period among younger adults in higher education institutions and workplaces. Thus, our results may not be externally generalised to the whole population. Nevertheless, we should emphasise that the findings represent the actual real-life situation in the field.

Thirdly, the small sample size for some associated factors was small, and this could cause insufficient study power and wide confidence intervals. Fourthly, our long COVID surveillance by telephone call might not have adequately excluded the alternative explanations for some symptoms and mistakenly assigned them as long COVID. This was unavoidable as our medical doctors could not examine them physically and perform relevant investigations. Nevertheless, we referred them all to government clinics for further management. Finally, we could not follow up with patients further to observe the persistence of long COVID symptoms beyond twelve weeks and the outcomes of our referrals.

## CONCLUSIONS

In conclusion, more than a quarter of our COVID-19 patients in Port Dickson, Malaysia, experienced long COVID symptoms. Females, underlying CVD, asthma and COAD, symptomatic care, myalgia, and headaches at COVID-19 diagnosis were independent factors associated with long COVID. Long-term surveillance and management of long COVID should be considered by policymakers, public health practitioners, and clinicians as we transition to living with COVID-19. Attention can be given to long COVID high-risk groups for targeted intervention. Our field experience showed that long COVID active case detection is feasible using telephone interviews. Sufficient monetary, material, and human resources are needed to implement this initiative.

## ACKNOWLEDGEMENTS

The authors would like to thank the Director-General of Health Malaysia for his permission to publish this article. We applaud all Port Dickson District Health Office staff for their support and dedication during this unprecedented COVID-19 pandemic.

### Funding
The authors received no funding for this work.

### Competing Interests
The authors declare there are no competing interests.

### Author Contributions
- Kim Sui Wan conceived and designed the experiments, performed the experiments, analyzed the data, prepared figures and/or tables, authored or reviewed drafts of the article, and approved the final draft.

- Esther Rishma Sundram conceived and designed the experiments, performed the experiments, prepared figures and/or tables, authored or reviewed drafts of the article, and approved the final draft.
- Ammar Amsyar Abdul Haddi performed the experiments, authored or reviewed drafts of the article, and approved the final draft.
- Abdul Rahman Dashuki performed the experiments, authored or reviewed drafts of the article, and approved the final draft.
- Azainorsuzila Ahad performed the experiments, authored or reviewed drafts of the article, and approved the final draft.
- Rowena John performed the experiments, authored or reviewed drafts of the article, and approved the final draft.
- Muhammad Khairul Ridhuan Abdul Wahid performed the experiments, authored or reviewed drafts of the article, and approved the final draft.
- Ungku Izmin Farah Ungku Halmie performed the experiments, authored or reviewed drafts of the article, and approved the final draft.
- Farah Edura Ibrahim performed the experiments, authored or reviewed drafts of the article, and approved the final draft.
- Nachia Banu Abdul Rahim performed the experiments, authored or reviewed drafts of the article, and approved the final draft.

### Human Ethics

The following information was supplied relating to ethical approvals (i.e., approving body and any reference numbers):

The Medical Research Ethics Committee (MREC) Ministry of Health Malaysia approved this study (NMRR ID-22-01103-GRF).

### Data Availability

The dataset retrieved and analysed in this study belongs to the Ministry of Health Malaysia. It is not available publicly due to local regulations imposed by the Medical Review and Ethics Committee (MREC), Ministry of Health Malaysia. Thus, raw data could not be provided or deposited in an open-access platform. However, a request for the dataset could be obtained via a written formal letter of application to the Director-General of Health, Malaysia https://www.moh.gov.my/index.php/edirectory/member_list/17/1.

### Supplemental Information

Supplemental information for this article can be found online at http://dx.doi.org/10.7717/peerj.14742#supplemental-information.

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
