# Peer review of "Long COVID active case detection initiative among COVID-19 patients in Port Dickson, Malaysia: a retrospective study on the positive outcomes, the proportion of patients with long COVID and its associated factors"

_PeerJ, doi:10.7717/peerj.14742_

## Round 0.1 · original submission · Minor Revisions

Dear Author,
Overall, the manuscript is well managed.

- In the abstract, I suggest adding the gap or problem statement in the background. The sampling technique should be mentioned and please clarify the multivariate logistic regression, is it binary or other? Please provide the study implication in the conclusion sections.

- In the introduction, overall is clearly stated. However, I suggest you provide a short explanation of the theoretical framework related to the problem or gaps in your study.

- In the method sections, please mention the reason why do you choose the Port Dickson District as your study location? I suggest you break down the data collection and present the information about independent and dependent variables clearly. You also need the present the instrument used clearly. Providing references would be better.

- In the discussion section. it would be better if you did not re-write the percentage or any statistical values from the findings. You should present the meaning of it and discuss it.

- Conclusion, please make a specific related to the program that you initiate to the policymakers and etc.

In the end, please read the guideline carefully and follow each instruction such as the way to write the reference properly. I saw many in-consistency in the reference style.

Reviewer 1 ·

Basic reporting

The authors have written and explained the facts and problems well. So, the readers can figure out what is the problem and aim of this study. But your introduction need to strengthen the novelty of the study. The previous study about long COVID and it's associated factors in Malaysia had done (https://doi.org/10.1101/2022.03.09.22272168). Please explain what is the difference of your study with that previous study. and it is better if the author also mentions the other previous study out of what I have mentioned before.

Experimental design

The authors have explained the methods well. But I have to clarify:
1. Line 105: What does the author mean about 1988?
2. Line 109: the authors have classified obesity based on the body weight is more than 90 kg. please give the rational or reference. And why the authors did not use other classification system? The body mass index classification method for example.

Validity of the findings

The author has presented and explained the novelty of the result by including the facts, rationale, and opinions in the discussion section well. But I want to clarify:
1. Line 233-234: to strengthen this novel finding, I suggest the author to review the pre-existing conditions CVD that has the potential for exacerbation of the severity of COVID.
2. Line 295-297 about mental health issues. What are the factors that make long COVID patients have higher risk for mental health problem?

Additional comments

no comment

·

Basic reporting

This article is already good enough, it just needs a slight improvement on some of the sections that I have already marked in the file

Experimental design

a. Why the authors choose in Port Dickson? Is it because of the highest number of long COVID cases in Malaysia, or other reasons, please explain

b. How to get the sample size?

Validity of the findings

c. Please add the variable section, which consists of dependent and independent variables of this study, a tool used to assess each variable and how it is calculated, as well as determine the value of its validity and reliability

Additional comments

Little grammar and spell check needs to be done.

---

## Round 0.2 · accepted · Accept

All reviewers' and editor's comments were addressed.